# Breeding Value Estimation Based on Morphological Evaluation of the Maremmano Horse Population through Factor Analysis

**DOI:** 10.3390/ani14152232

**Published:** 2024-07-31

**Authors:** Andrea Giontella, Maurizio Silvestrelli, Alessandro Cocciolone, Camillo Pieramati, Francesca Maria Sarti

**Affiliations:** 1Department of Veterinary Medicine—Sportive Horse Research Center, University of Perugia, Via S. Costanzo 4, 06126 Perugia, Italy; maurizio.silvestrelli@unipg.it (M.S.); camillo.pieramati@unipg.it (C.P.); 2Department of Agricultural, Food and Environmental Sciences, University of Perugia, Borgo XX Giugno, 74, 06121 Perugia, Italy; alessandro.cocciolone@unipg.it (A.C.); francesca.sarti@unipg.it (F.M.S.)

**Keywords:** continuous evaluation scale, genetic parameters, breeding value, optimized selection strategies, Maremmano horse

## Abstract

**Simple Summary:**

Horse body conformation is an important aspect to evaluate in sport horse selection since the overall body shape determines the limits of the range of movement and function of the horse and, ultimately, its future performance. The genetic improvement of this breed is under the control of the National Association of Maremmano Breeders (ANAM), the first Italian association to use a continuous evaluation scale to evaluate the horses enrolled in the Studbook in the early 1990s. Morphological scoring is a common evaluation method for domestic animals. People employed in this type of role must be able to translate their judgements regarding certain features of the animal’s body into numbers or letters, thus avoiding the need to measure the animal. Only a few aspects of the horse’s body conformation are actually measured since this activity is difficult and expensive. This study proposes an easier approach to horse scoring in order to obtain genetic proof of morphological traits and, consequently, to support breeders in making targeted selection decisions. Three new factors were computed and summed through an unique New Estimated Breeding Value (NEBV1), with the purpose of supporting breeders in adopting an easier and more accurate selection approach.

**Abstract:**

Morphological scoring is a common evaluation method for domestic animals. The National Association of Maremmano Breeders (ANAM) has provided a dataset containing the records of 600 horses, four metric measurements (cm) and 24 traits with a continuous evaluation scale, each one with 15 classes. Moreover, a body condition score (BCS) with five classes is included. In this study, factor analysis was conducted to create a small number of informative factors (3) obtained from these traits, and a new BLUP-AM-MT index was established. The New Estimated Breeding Value (NEBV1) of each horse was computed by adding the genetic indexes of the three factors, with each one multiplied using a coefficient indicated by ANAM. The practical feasibility of the NEBV1 was evaluated through Spearman correlations between the rankings of the NEBV1 and the rankings of the BLUP-AM-MT, estimated through the four biometric measures and the morphological score (MS) assigned to each horse by the ANAM judges. The factorial analysis was used to estimate three factors: the “Trunk Dimension”, “Legs” and “Length”. As the explained variance was only 32%, the model was rotated, and the heritability of the three factors were 0.51, 0.05 and 0.41, respectively. After rotation, the estimated correlations between the new NEBV1 and the biometric measures were improved. These results should encourage breeders to adopt a breeding value index that takes into consideration the factors derived from all the variables observed in the morphological evaluation of the Maremmano. In this way, breeders can use it to select the best animals for breeding.

## 1. Introduction

### 1.1. The Morphological Scoring

Morphological scoring is a common evaluation method for domestic animals. People employed in this type of role must be able to translate their judgements regarding certain features of the animal’s body, comparing it to an ideal type described in the breed standard, into numbers or letters [1]. Historically, most horse breeders have followed a selection based only on esthetic standards, but in recent decades, more emphasis has been placed on functional conformation [2,3,4]. Conformation is, indeed, an important feature of horse breeding, as the overall body shape determines the limits of the range of movement and function of the horse and, ultimately, its future performance (ability to perform) [5,6]. In the case of the sport horse, the indirect selection of performance traits (e.g., flat race, trot, dressage, show-jumping) using information on conformation has proven to be useful. In fact, performance traits have a low heritability (h^2^) [7,8,9,10] and can only be measured late in the horse’s life. Only in the Thoroughbred do the heritability estimates for racing range widely, varying from nearly zero to upward of 0.75, as reported by Raudsepp et al. (2019) [11]. On the contrary, conformation traits can be recorded earlier [5,12]. The efficiency of indirect selection for performance depends on the genetic variation in the conformation traits and the genetic correlations between the conformation traits and performance traits [13,14]. A large number of objective conformation studies based on body measurements have been carried out, usually to evaluate the heritability of the traits [15,16,17,18,19,20].

Despite this, in the case of Italian breeds, only a few traits of the horse’s body conformation are actually measured since this activity is difficult and expensive. Morphological scoring was once widely employed. This method compares the horse to an “ideal type”, generally during conformation and gait shows [1]. Conformation traits that were mostly classified on a subjective scale showed low to moderate heritability [21,22]. Moreover, only moderate genetic correlations between movement traits and the results of competitions were identified [23,24,25]. The main problem with this procedure is its subjectivity [26]. Subjective judging is defined as judging through the use of individual feeling or apprehension as the ultimate criterion with respect to what is deemed good and correct [27]. In fact, it is difficult to determine whether different judges, although well trained, offer the same assessment of the same animal [26]. For this reason, morphological scoring has been progressively replaced by the linear scoring system, which is based on the objective measure of the body traits and not on attempts to define good or bad characteristics. Rather, it describes where the individual being assessed lies on a scale between the biological extremes of a particular conformation trait.

### 1.2. The Linear Scoring System

The concept of linear scoring was conceived in 1976 and was first applied and tested in 1979 [28]. In Italy, linear scoring was firstly applied to dairy cattle, and a number of years later, the beef cattle associations also established linear scoring protocols [29]. In addition to cattle, linear scoring is also used for water buffalo and for sheep, even if the short biological interval and, consequently, the limited range of each class render correct scoring difficult in the latter species [26].

Likewise, linear scoring is used for several horse breeds in Italy; the National Association of Maremmano Breeders (ANAM) was among the first to use a continuous evaluation scale (a preliminary evaluation for the application of linear scoring) to evaluate horses destined to be added to the Studbook, and it is still used for the selection of the breed. Maremmano is an indigenous Italian horse with a well-established phenotype, despite a long history of crossbreeding. Over the course of history, the breed has been crossed with Arab, Barb and Spanish Purebred horses. Today, all Maremmano horses are descended from a greatly limited number of male lines, and each living horse is officially assigned to one of four lines [30]. It is evident that multiple dam lineages have contributed to the extant genetic and morphological variability [31]. The establishment of the Studbook occurred in 1980, and a conservation program for the breed started in 1992. The Maremmano horse was the first Italian “warmblood” to be subjected to an experimental evaluation of young stallions through the 100-day stationary performance test, with morphological (continuous), clinical and performance evaluations [32,33]. With the wider implementation of this continuous evaluation scale of conformation and performance, accurate phenotypic information became routinely available for this breed. This study aimed to develop a simpler and less biased scoring method through a new statistical assessment of the data for the breeding value estimation of morphological traits.

## 2. Material and Methods

### 2.1. Animals

The study was carried out on a dataset including 600 horses (40 males, 560 females) that was provided by the National Association of Maremmano Breeders (ANAM) (Appendix A). For each animal, the analyzed variables are reported in Table 1. The anatomical parts from which they were collected are shown in Figure 1 and Appendix A.

Four variables were expressed in centimeters (*biometric measurements*), while the others were evaluations (continuous evaluation scale) expressed in 15 score classes. Moreover, the body condition score (BCS) was evaluated in 5 classes (1—minimum to 5—maximum) by a number of judges during the morphological evaluation, and the biometric measurement was considered. To better explain the aim of the continuous evaluation of the Maremmano horse, following the method in a paper on Pura Raza Español horses [18], the traits were divided into four categories. The first referred to the traits that describe the *body proportional balance*, the second referred to the traits related to *sport performance*, the third referred to the traits related to *leg structure correctness* and the fourth to the *evaluation of body fat*. In a preliminary screening of the dataset, it was observed that continuous evaluations were carried out by 25 judges. Since some of them provided a very low number of judgements (<5), they were not considered. Therefore, only 15 judges remained in the dataset. Finally, each horse was evaluated once by the given judge.

### 2.2. Preliminary Statistical Analyses

The correlations between the scores of each judge and the objective metric measures of the CG and SLL were also estimated to verify the judges’ scoring ability. All the judges with a correlation lower than 0.30 were eliminated (see Appendix A). The final dataset contained the scores provided by 11 judges for 501 horses (40 stallions and 461 mares). In the scoring form, all the scores were expressed as letters (15 scores from “A” to “Q”) following the Italian alphabet. Therefore, a transformation from letters to numbers was carried out before the statistical analysis. For continuous quantitative traits (HPL, NLL, SPL, WHL, CWL, CHL, BLL, LLL, CLL and CrWL), the value of 1 was assigned to the lowest class (class A in the scoring form), and the values of the other classes extended up to 15 in the case of the highest class (class Q in the scoring form). For continuous traits related to formal body correctness and the typical characteristics of the breed (HVL, NAL, LBL, SCL and all the traits related to the leg structure), the value of 15 was assigned to the more desirable central class (horses without morphological defects and very typical in their traits; H in the scoring form), and the two sides were considered as two traits, decreasing symmetrically in steps of two. Before the statistical analysis, the age on the date of evaluation was computed, and the obtained values were grouped into 3 classes, as reported in the literature [34], namely: CLASS 1 (young animals): 2–4 years (166 horses), CLASS 2 (adult animals): 5–15 years (261 horses), and CLASS 3 (mature animals): >15 years (74 horses). In a preliminary data analysis, the frequencies of each class and the mode for all the continuous traits, as well as the mean, the standard error and the coefficient of variation (CV) of the biometric measures and all continuous traits, were estimated for the total sample and by sex. The Shapiro–Wilk normality test was performed to determine whether the available data were well modeled by a normal distribution, and the homogeneity of variance was verified using Bartlett’s test. Statistical analysis was carried out using the R software environment (ver. 4.2.2) [35].

### 2.3. Factor Analysis

Because of their large number, it was difficult to synthesize all the variables into only one BLUP-AM “Breeding Value”. For this purpose, a factor analysis (PROC FACTOR, SAS) was carried out to obtain a small number of informative factors and to compute a unique, suitable BLUP-AM-MT index. In the first factor analysis, only the continuous evaluation scale variables were considered, while in the second factor analysis, all the variables (biometric and continuous) were examined. The variables useful for the estimation of the BLUP-AM index were selected according to the Kaiser–Meyer–Olkin (KMO) test, which determines how suitable data are for factor analysis. Moreover, the factorial axes were rotated by the VARIMAX method to minimize the number of variables that had strong correlations with the factors. The number of factors was fixed at 3.

### 2.4. Animal Model

The genetic indexes of the 3 factors were estimated through a mixed-model analysis that took into account the sex, judge, BCS and age class as fixed effects, as defined by BLUPF90 [36], according to an individual animal model that considers a relationship matrix of 16,988 animals (with 17 traced generations and a pedigree completeness of 99%). The New Estimated Breeding Value (NEBV1) was computed by adding the genetic indexes of the three factors, with each one multiplied using the coefficient indicated by ANAM:NEBV1 = (0.4 × Factor 1) + (0.3 × Factor 2) + (0.3 × Factor 3).

The practical feasibility of the NEBV1 was evaluated through Spearman correlations between the rankings of the NEBV1 and the BLUP-AM-MT, estimated through the four biometric measures and the morphological score (MS) assigned to each horse by the ANAM judges. The BLUP-AM-MT and the morphological scores (MS) were obtained from the ANAM archives.

## 3. Results

The variables were not normally distributed (*p* < 0.05), and the homogeneity of the variances was confirmed (*p* > 0.05). In the sample, the average age of the horses at the date of evaluation was 9 ± 5.7 years. In Table 2, Table 3 and Table 4, the frequencies of the continuous evaluation scale classes are reported, respectively, for the body proportional balance, sport performance traits and leg structure correctness.

As expected, the frequencies reported in Table 2 have a unimodal distribution, where the central class (H) is always the most abundant one, as in the case of the LBL (64%), HPL (48%) and NAL (44%) traits. The only difference is the SPL, where the H class has a 15% frequency, and the most abundant class is F (23%). The extreme classes (A, B, P, Q) have no observations, and the others show greater values moving closer to the central one.

The above-described trend is less evident in the sport performance traits (Table 3), where the central class is often not the most abundant one (SLL, CWL, CHL, LLL), and the highest values are observed in the classes greater than the central one. This situation could be due to the selection target, which prefers a rather light type of horse.

The values reported in Table 4 usually show no observations in the first and last classes. The central class (H) is always well represented, and sometimes, its value is over 80% (FFKK, FSCK and HLCH). This situation is related to the type of trait (structure of the legs), in which the central class represents the animals with correct leg structure, and the other classes represent the horses with severe defects. The traits with the highest defect frequencies are the FFBN, FFTO, HLBN, HLTO and PAL.

The means, standard errors and coefficients of variation reported in Table 5 show the relatively large dimensions of the sampled horses. For the biometric measures, in the total sample, the height of the withers is 164.31 ± 0.20 cm and is highly proportional to the large chest circumference of 194.11 ± 0.56 cm. The cannon bone circumference is 20.78 ± 0.06 cm, and the shoulder length is 69.66 ± 0.24 cm. The means determined by sex are not different from the overall means. The sample is rather homogeneous; the standard errors are small, and the coefficients of variation are very low, ranging from 2.7% (WH) to 7.80% (SL).

The mean ± SE of the body proportional balance traits ranges from a minimum of 7.35 ± 0.08 in the SCL to a maximum of 9.15 ± 0.08 in the WHL, and the most common class ranges from 7 to 9. The sport performance traits range between 7.98 ± 0.07 in the NLL and 9.71 ± 0.07 in the SLL, and the mode ranges between 8 and 10. The leg structure correctness traits range from 7.32 ± 0.06 in the HLBN to 8.19 ± 0.07 in the FFTO, and the mode is equal to 8 for all these traits. The value for the BCS is 3.35 ± 0.03 (mode = 3), and the coefficients of variation are higher with respect to the biometric measures, ranging between 8.59% (FFKK) and 28.95% (SCL).

In the first factorial analysis, Factor 1 was well correlated with the CWL (0.75), CHL (0.80) and CrWL (0.76) and, therefore, is called the *Trunk Dimension*. Factor 2 showed rather strong correlations with the FFKK (0.54) and HLCH (0.58) and, thus, is called the *Legs Factor*. Factor 3 was correlated with the NAL (−0.67), BLL (0.60) and LLL (0.78) and is called the *Length Factor*. The result of the KMO test in this analysis was low (0.64), and the explained variance was equal to 32%.

To verify the suitability of these three factors to be used in Maremmano selection, their heritability was estimated. The h^2^ values were moderate for the first two factors (0.31 and 0.36, respectively), but low in the case of the third factor (0.14) (Table 6); through these three factors, a BLUP-AM-MT index was estimated. Each of the three factors was multiplied by a specific coefficient and then summed to obtain the New Estimated Breeding Value (NEBV). The weighting coefficients, defined together with the supervisors of the ANAM, were 0.4 for Factor 1 and 0.3 for Factors 2 and 3. The Spearman rank correlation computed between the NEBV and the BLUP-AM-MT indexes, which were estimated based on the four biometric traits (WH, CG, SL, BC), were low, ranging between 0.16 (NEBV-WH, NEBV-BC) and 0.20 (NEBV-CG). Moreover, it must be highlighted that the biometric indexes were strongly correlated with each other, and that these correlations ranged between 0.65 (WH-BC) and 0.85 (WH-SL). Special attention should be given to the correlations between the morphological scores and the BLUP-AM-MT indexes discussed above. The correlation coefficients for the biometric indexes were very low and ranged between 0.06 (MS-BC) and 0.17 (MS-WH). In addition, the correlation between the NEBV and the MS was very low (−0.02) and not significant (data not reported).

To overcome this problem, another BLUP-AM-MT index (NEBV1) was estimated, and a second factor analysis was then carried out on all the variables reported in Table 1. As shown in Figure 2, the first factor was always associated with the *dimension traits*, but in this case, the traits correlated with both the continuous evaluations (CWL, SLL, CHL, CrWL) and the biometric measures (WH, CG, BC, SL). The coefficients ranged between 0.46 (WH) and 0.65 (CHL). The second factor was similar to that estimated in the first factor analysis and was associated with the *leg traits* (FFKK, FSCK, HLTO, HLSH). The third factor represented the *length of the animal*, but in regard to the new analysis, we must highlight the negative correlation coefficient of the NAL (−0.67), while for the other two traits (BLL and LLL), the coefficients were 0.52 and 0.75, respectively. The KMO test result was slightly higher (0.66) than in the previous analysis, and the variance explained was 21%.

The new heritability coefficients and the genetic correlations are reported in Table 6. In the new analysis, the h^2^ value increased for Factors 1 and 3 (0.51 and 0.41), but for Factor 2, it was reduced (0.05), outlining the problems in selecting for non-quantitative traits. The genetic correlations between the 3 factors are all positive and medium–high.

The Spearman correlations estimated between the recalculated BLUP-AM-MT, named NEBV1 (Table 7) and the biometric measures were lower and significant, ranging between 0.41 (NEBV1-WH) and 0.20 (NEBV1-CG). The correlation between the NEBV1 and the morphological score (MS) was still low (0.12) and not significant.

## 4. Discussion

The results reported above reveal that the extreme classes of the continuous evaluation scale were seldom utilized by the judges. In this scoring system, the presence of empty classes does not affect the results, because the breeders spontaneously discard horses that, at the exhibition, would fall into the extreme classes. Furthermore, a possible reduction in the number of classes from 15 to 13 may be recommended, but this decision falls within the remit of the ANAM Technical Committee.

An important aspect of improving the continuous evaluation scale of the Maremmano is to ensure the systematic supply of new information and annual training of the judges, which should lead to the better application of the scoring scale of continuous profiling. In addition, in the training phase of the judges, it would be useful to have each horse evaluated several times by the same judge in order to test the repeatability and reproducibility of the judgements.

As reported by Sánchez-Guerrero et al. (2017) [18], whatever the purpose of the horse may be, the objective evaluation of its conformation and its relation to high sport performance is of great importance. Rooney et al. (1998) [37] reported that short cannons are desirable for any high-performance horse, as this reduces the weight of the lower leg so that less muscular effort is required to move the limbs, thus maximizing the jumping ability, while a long foreleg is best for speed, jumping and long-distance riding. In the case of the Thoroughbred, the biometric measure of the WH reported by Mawdsley et al. (1996) [38] ranged from 155.30 to 166.20, while in Senna et al. (2015) [39], the SL of the same breed was 69.00 cm. In this study, the results in Table 5 show that the means of these two traits in the Maremmano are very similar, indicating a good sporting aptitude. Moreover, in terms of the biometric measures, the studied sample is rather homogeneous. The standard errors are small, and the coefficients of variation are very low, ranging from 2.7% (WH) to 7.8% (SL). The means of these traits reveal that, in this breed, sexual dimorphism is not strong. In terms of the continuous traits, the population shows good variability for selection. In fact, the values of the CV are very similar to the values reported by Perdomo-González et al. (2022) [5] for 46 linear morpho-functional traits evaluated on 333 Raza Menorquina horses. The authors confirmed that the CV can be considered the most important measure of variation, and it generally assumes that the higher the phenotypic variation in the traits is, the greater the genetic variation will be, which guarantees a sufficient selection response in the population.

Cervantes et al. (2009) [40] studied 37 morphological traits using a multivariate analysis approach based on 171 Spanish Arab horses with an age range from 3 to 14 years. In the paper, the first factor in the principal component analysis explained 21.36% of the total variance in the grouped *height traits*, which represent the general size of the horse, and is very similar to the first factor in our analysis named the *trunk length*. The second factor included the *thoracic variables* (8.52% of the total variance), while the *angles related to functionality traits* were represented by the second and third factors (6.15% of the total variance), accounting for 36% of the total explained variance. These results are very similar to the values of the first factorial analysis (32%) in the present study. Moreover, in our case, the introduction of the biometric scoring variables into the second model resulted in a decrease in the overall variance explained by the model from 32% to 21%. In addition, the trait neck alignment (NAL) was assigned to the third factor, but the value of its correlation was negative. This factor (called *length*) is positively correlated with the back and loin length. This fact is very important, because it indicates that if the two lengths increase, the neck alignment will deviate from the optimal score reported in the form (Appendix A). In this context, another important element for discussion is the appropriate assessment for each trait of the form. It is not easy to objectively assess certain traits in the continuous evaluation, despite the fact that the evaluators follow specific stages and training courses. Regarding the genetic parameters, as previously reported, in the second factor analysis, the heritability coefficients for Factors 1 and 3 were 0.51 and 0.41, respectively, but in the case of Factor 2, it was reduced to 0.05. The value of the h^2^ of the second factor was similar to the heritability values estimated by Sánchez-Guerrero et al. (2017) [18] through multivariate analysis based on 13 morphological linear traits of Pura Raza Español horses. These values ranged from 0.12 for the ischium–stifle distance trait to 0.53 for the cannon bone perimeter. The magnitude of the second factor’s heritability was higher in the first factorial analysis, even though it was expected to be lower because it was related to the non-quantitative traits (Table 6). The genetic correlations between the three factors reported in Table 6 are very interesting. As is well known, in a multiple-trait index, the traits with a low heritability coefficient, as reported by Solé et al. (2014) [34], Sánchez-Guerrero et al. (2017) [18] and Perdomo-González et al. (2022) [5] (in this case, Factor 2), benefit strongly from the high heritability of the other two factors in the NEBV1 estimation, but mainly benefit from the strong genetic correlations between the others (Factor 2–Factor 1: 0.55/Factor 2–Factor 3: 0.99). Special attention must be given to the correlations between the morphological scores and the BLUP-AM indexes discussed above.

The estimated heritability of the traits in the Maremmano horse breed reveals that genetic improvement is possible, and this study allows us to foresee the possibility of making good genetic progress through the breeding program; this fact is also supported by the moderate variability calculated for the continuous traits.

As pointed out by Perdomo-González et al., 2022 [5], understanding the relationships between morphological traits is extremely useful in animal breeding for the determination of both the breeding criteria and the possible breeding response in selection programs.

The correlation coefficients with the biometric indexes are very low, ranging between 0.06 (MS-BC) and 0.16 (MS-CG); moreover, the correlation between the NEBV1 and MS is very low (0.12) and not significant (Table 7). This situation causes confusion in selection planning, because there are ambiguous ways of selecting the animals to mate. However, the Spearman rank correlations between the measurable variables (WH, CG, BC, SL) ranged between 0.61 and 0.85, which could be useful in reducing the number of traits to be measured. For instance, the correlation between the WH and SL is positive and high (0.85), and such a situation might lead to the decision to select only the easiest trait to measure (for example, the WH), rendering the work of the experts less expensive. Therefore, the correlations between the continuous traits must be estimated before including morpho-functional traits in the selection and mating program of a given breed [5].

Many attempts have been made in order to simplify the collection of measurable traits. Gaudioso et al. (2014) [41] studied a photozoometer that allows one to obtain measurements using 3D image analysis, focusing on the Alpine Brown and Friesian cattle breeds. Pallottino et al. (2015) [42] tested a stereovision system on the Lipizzan horse, and in 2020, Pérez-Ruiz et al. [43] carried out tests using a digital three-dimensional model using LiDAR on the Pura Raza Española horse. In the future, these advanced techniques might offer the possibility of collecting more measurable traits, which are required in order to calculate more accurate EBVs and to avoid the excessive time and labor of experts involved in this activity. Until then, the use of the continuous evaluation scale and measurable traits simplified through the factorial approach may offer an alternative method that can be used to enhance the efficiency of genetic selection for the Maremmano horse. The results of the present work should encourage breeders to use a breeding value index that takes into consideration the traits (factors) derived from a continuous evaluation observed through the morphological evaluation of this breed.

## 5. Conclusions

Nowadays, despite the large-scale dissemination of genetic evaluation techniques, morphology has maintained its importance since several of the characteristics surveyed have significant impacts on the different selection objectives and are essential for QTL analyses. The transition from a continuous evaluation scale to linear evaluation is of fundamental importance when evaluating totipotent horses such as the Maremmano horse, because it is the only method used for the selection process. However, as this study shows, the most important problem to solve is the objective compilation of the morphological evaluation form. It is, therefore, essential to continue to plan training courses for experts and to stimulate discussion with the breeding associations in order to provide an objective linear evaluation independent of the judge. An appropriate assessment of the Maremmano horse based on morphological conformation is one of the crucial stages in the development of strategies and plans for genetic improvement that meets the current needs of the market, whether the horses are sport or work mounts. The results could be used to better redefine the current approach used to collect linear morphological data on the Maremmano breed since these data have proved to be the basis of the indirect genetic selection of many traits.

The heritabilities and correlations estimated in this study suggest that a selection program can be based on the genetic evaluation of linear profiling, even though the evaluated database is not quite optimal from the perspective of data quality. The genetic evaluation of linear profiling through this new statistical approach will lead to improvements in the conformation traits according to the defined breeding objectives. The prediction of new breeding values should be the next step following this study, which will allow for the development of more effective selection plans in regard to the conformation and use of a horse.

Finally, this research could be useful for delineating a more accurate selection approach. Farmers could use the new NEBV1 to select the best animals suited to the ideal type for mating plans.

## Figures and Tables

**Figure 1 animals-14-02232-f001:**
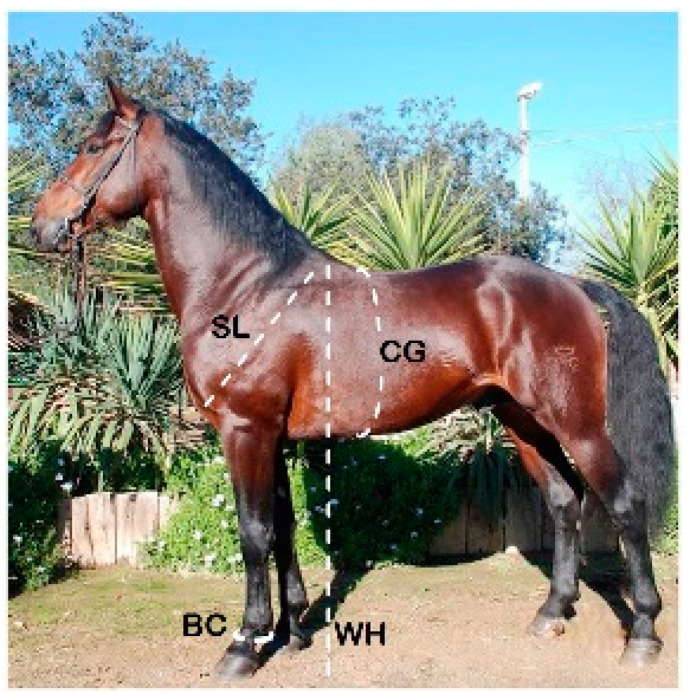
The four body measurements detected in the Maremmano horse. WH = wither height; CG = chest girth; BC = cannon bone circumference; SL = shoulder length.

**Figure 2 animals-14-02232-f002:**
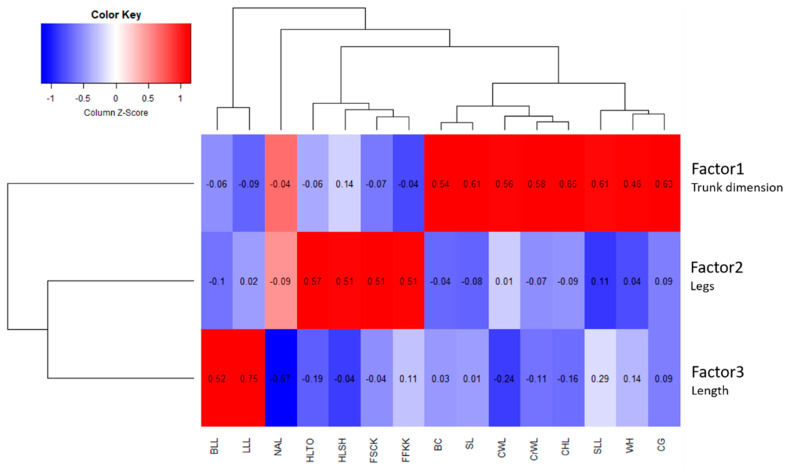
Second factor analysis results. WH = wither height; CG = chest girth; BC = cannon bone circumference; SL = shoulder length; CWL = chest width; SLL = shoulder length; CHL = chest height; CrWL = croup width; FFKK = foreleg—front view: knock-kneed; FSCK = foreleg—side view: calf-kneed; HLTO = hind leg—rear view: toes out; HLSH = hind leg—side view: sickle-hocked; NAL = neck alignment; BLL = back Length; LLL = loin length.

**Table 1 animals-14-02232-t001:** General data, variables and related acronyms.

General Data	Acronym
Studbook number	ID
Card number	NUM
Judge	JDG
Evaluation date	DATE
Judgement location	LOC
**Variable**	
*Biometric Measurements*	
Wither height (cm)	WH
Chest girth (cm)	CG
Cannon bone circumference (cm)	BC
Shoulder length (cm)	SL
**Continuous Evaluation Scale**	
*Body proportional balance*	
Head profile	HPL
Head volume	HVL
Neck alignment	NAL
Shoulder position	SPL
Wither height	WHL
Line of the back	LBL
Shape of the croup	SCL
*Sport performance*	
Neck length	NLL
Chest width	CWL
Shoulder length	SLL
Chest height	CHL
Back length	BLL
Loin length	LLL
Croup length	CLL
Croup width	CrWL
*Legs structure correctness*	
Foreleg front view: base narrow	FFBN
Foreleg front view: toes out	FFTO
Foreleg front view: knock-kneed	FFKK
Foreleg side view: calf-kneed	FSCK
Hind leg-rear view: base narrow	HLBN
Hind leg-rear view: toes out	HLTO
Hind leg-rear view: cow-hocked	HLCH
Hind leg-side view: sickle-hocked	HLSH
Pastern angle	PAL
*Evaluation of Body Fat*	
Body condition score	BCS

**Table 2 animals-14-02232-t002:** Frequencies (%) of body proportional balance classes (n = 501).

Traits	Frequencies (%)															
	A	B	C	D	E	F	G	H	I	L	M	N	O	P	Q	Total
HPL	-	-	-	1	2	3	8	47	14	15	7	2	1	-	-	100
HVL	-	-	1	2	5	11	10	35	11	18	5	2	-	-	-	100
NAL	-	-	-	1	4	9	12	44	6	11	8	3	2	-	-	100
SPL	-	-	2	4	14	23	13	15	8	14	6	1	-	-	-	100
WHL	-	1	1	2	7	6	23	11	22	16	6	4	1	-	-	100
LBL	-	-	1	2	3	8	12	64	6	3	1	-	-	-	-	100
SCL	-	-	-	1	4	6	5	26	14	22	16	4	2	-	-	100

HPL = head profile; HVL= head volume; NAL =neck alignment; SPL = shoulder position; WHL = wither height; LBL = line of the back; SCL = shape of the croup.

**Table 3 animals-14-02232-t003:** Frequencies (%) of sport performance trait classes (n = 501).

Traits	Frequencies (%)															
	A	B	C	D	E	F	G	H	I	L	M	N	O	P	Q	Total
NLL	-	-	-	2	7	17	13	26	8	16	9	2	-	-	-	100
CWL	-	-	1	3	5	9	8	19	8	21	16	5	4	1	-	100
SLL	-	-	-	1	1	4	3	17	11	27	25	8	3	-	-	100
CHL	-	-	-	1	2	5	6	19	13	22	19	7	5	1	-	100
BLL	-	-	-	1	1	4	5	43	16	17	9	2	1	1	-	100
LLL	-	-	-	-	2	5	7	21	21	25	12	4	2	1	-	100
CLL	-	-	1	1	7	7	12	34	11	16	9	1	1	-	-	100
CrWL	-	-	-	-	2	3	4	27	12	26	17	5	4	-	-	100

NLL = neck length; CWL = chest width; SLL = shoulder length; CHL = chest height; BLL = back length; LLL = loin length; CLL = croup length; CrWL = croup width.

**Table 4 animals-14-02232-t004:** Frequencies (%) of leg structure trait classes (n = 501).

Traits	Frequencies (%)															
	A	B	C	D	E	F	G	H	I	L	M	N	O	P	Q	Total
FFBN	-	-	1	2	5	9	13	52	8	6	2	1	1	-	-	100
FFTO	-	-	1	1	5	9	8	44	10	11	7	2	1	1	-	100
FFKK	-	-	-	-	1	2	7	83	4	2	1	-	-	-	-	100
FSCK	-	-	-	-	1	3	6	84	3	2	1	-	-	-	-	100
HLBN	-	-	1	2	9	18	16	40	9	4	1	-	-	-	-	100
HLTO	-	-	1	1	5	18	16	53	2	3	1	-	-	-	-	100
HLCH	-	-	-	-	1	2	9	81	4	2	1	-	-	-	-	100
HLSH	-	-	1	1	2	5	10	56	11	9	4	1	-	-	-	100
PAL	-	-	1	2	7	16	14	44	7	7	2	-	-	-	-	100

FFBN = foreleg front view: base narrow; FFTO = foreleg front view: toes out; FFKK = foreleg front view: knock-kneed; FSCK = foreleg side view: calf-kneed; HLBN = hind leg-rear view: base narrow; HLTO = hind leg—rear view: toes out; HLCH = hind leg–rear view: cow-hocked; HLSH = hind leg–side view: sickle-hocked; PAL = pastern angle.

**Table 5 animals-14-02232-t005:** Descriptive statistics for the biometric measures (overall and by sex) and for the continuous evaluation traits.

	Overall		Males		Females	
Trait	Mean ± SE	CV	Mean ± SE	CV	Mean ± SE	CV
	(cm)	%	(cm)	%	(cm)	%
WH	164.31 ± 0.20	2.70	164.62 ± 0.69	2.64	164.09 ± 0.21	2.74
CG	194.11 ± 0.56	6.45	194.46 ± 1.48	4.81	193.84 ± 0.67	7.42
BC	20.78 ± 0.06	6.21	20.81 ± 0.25	7.64	20.73 ± 0.05	4.82
SL	69.66 ± 0.24	7.80	69.86 ± 0.83	7.54	69.38 ± 0.26	7.98
			**Mean ± SE**		**Mode**	**CV**
*Body proportional balance*						
HPL			8.58 ± 0.05		8	16.45
HVL			8.15 ± 0.07		8	21.14
NAL			8.19 ± 0.07		8	21.96
SPL			8.98 ± 0.07		7	21.05
WHL			9.15 ± 0.08		9	22.62
LBL			7.67 ± 0.04		8	15.63
SCL			7.35 ± 0.08		9	28.95
*Sport performance*						
NLL			7.98 ± 0.07		8	24.37
CWL			8.90 ± 0.09		9	25.61
SLL			9.71 ± 0.07		10	17.96
CHL			9.47 ± 0.07		10	20.58
BLL			8.74 ± 0.06		8	16.84
LLL			9.11 ± 0.06		9	18.14
CLL			8.28 ± 0.07		8	22.65
CrWL			9.35 ± 0.07		10	18.84
*Leg structure correctness*						
FFBN			7.70 ± 0.06		8	19.36
FFTO			8.19 ± 0.07		8	22.38
FFKK			7.94 ± 0.02		8	8.59
FSCK			7.97 ± 0.02		8	9.02
HLBN			7.32 ± 0.06		8	20.32
HLTO			7.38 ± 0.04		8	16.56
HLCH			7.94 ± 0.03		8	9.33
HLSH			8.17 ± 0.05		8	16.48
PAL			7.48 ± 0.06		8	19.96
*Evaluation of Body Fat*						
BCS			3.35 ± 0.03		3	19.45

WH = wither height; CG = chest girth; BC = cannon bone circumference; SL = shoulder length; HPL = head profile; HVL = head volume; NAL = neck alignment; SPL = shoulder position; WHL = wither height; LBL = line of the back; SCL = shape of the croup; NLL = neck length; SLL = shoulder length; CWL = chest width; CHL = chest height; BLL = back length; LLL = loin length; CLL = croup length; CrWL = croup width; FFBN = foreleg front view: base narrow; FFTO = foreleg front view: toes out; FFKK = foreleg—front view: knock-kneed; FSCK = foreleg—side view: calf-kneed; HLBN = hind leg—rear view: base narrow; HLTO = hind leg—rear view: toes out; HLCH = hind leg—rear view: cow-hocked; HLSH = hind leg—side view: sickle-hocked; PAL = pastern angle; BCS = body condition score.

**Table 6 animals-14-02232-t006:** Heritability ± SE of the three factors of the two factorial analyses on the diagonal (first factorial analysis above; second factorial analysis below) and genetic correlations above the diagonal (first factorial analysis above; second factorial analysis below).

	Factor 1	Factor 2	Factor 3
Factor 1	**0.31 ± 0.04**	0.31 ± 0.03	0.26 ± 0.02
	**0.51 ± 0.03**	0.55 ± 0.01	0.57 ± 0.01
Factor 2		**0.36 ± 0.05**	0.57 ± 0.04
		**0.05 ± 0.04**	0.99 ± 0.03
Factor 3			**0.14 ± 0.03**
			**0.41 ± 0.01**

**Table 7 animals-14-02232-t007:** Spearman correlations between the estimated BLUP-AM-MT and morphological scores after the second factor analysis.

BLUP-AM-MT Indexes and MS	BLUP-AM-MT Indexes				
	NEBV1	WH	CG	SL	BC
WH	0.41 ***	-	-	-	-
CG	0.20 ***	0.78 ***	-	-	-
SL	0.40 ***	0.85 ***	0.82 ***	-	-
BC	0.32 ***	0.65 ***	0.70 ***	0.61 ***	-
MS	0.12 ns	0.10 **	0.16 **	0.10 *	0.06 ns

WH = wither height; CG = chest girth; BC = cannon bone circumference; SL = shoulder length; MS = morphological score; NEBV1 = New Individual Breeding Value. * *p* < 0.05, ** *p* < 0.01 and *** *p* < 0.001; ns = not significant.

## Data Availability

Data are contained within the article and Appendix A.

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
