# Peer review of "Breeding Value Estimation Based on Morphological Evaluation of the Maremmano Horse Population through Factor Analysis"

_animals, 2024, doi:10.3390/ani14152232_

Round 1

Reviewer 1 Report

Comments and Suggestions for Authors

The authors aimed to develop a simpler and less biased scoring method through a new statistical assessment of the data for the breeding value estimation of morphological parameters for Maremmano horse breed. They used factorial analysis was used to estimate three factors: the Trunk Dimension, Legs and Length. The authors argued that after rotation, the estimated correlations between the new BV and the biometric measures were improved and breeders can use it to select the best animals for breeding.

The “Introduction” section is too long. Please shorten the text.

The “Materials and Methods” section is enough and understandable. The supplementary document is adequate and informative.

The “Results” section is OK.

The “Discussion” section the authors mentioned that CV values were so small. Indeed it is the fact that your scores were between 1 and 15, it is expected. Your Spearman rank correlation values didn’t support your decision about the breeding value. I think that some other statistical methods, more informative, can be used.

The “Conclusion” section is compatible with results. But “Farmers could use the new IBV to select the best animals suited to the ideal type for mating plans.” I don’t agree with you in the light of the results. This model should be improved before offer o use. And you mentioned at line 69 that “The main problem with this procedure is its “subjectivity””. You are right.

This manuscript can be accepted after major revision.

1. Lines 21-22: “Individual Breeding Value (IBV)”, The Breeding Value is individual that is not need to write it. It should be “Breeding Value (BV)”

2. Line 190: “Shapiro–Wilk normality test was performed”. Please use Kolmogorov-Smirnow test because your data is over than 50 samples. And for homogeneity you used “Bartlett’s test”, why didn’t you use Levene test?

3. Line 216: “were not normally distributed (P > 0.05)”. If (P > 0.05), it means that the data can be assumed normally distributed. I think it is a miswriting.

4. Line 226: The extreme classes was started with B and finished with O. there is no data for some classes. You should mention it at methods section. No 15 classes.

5. Line 244: The sport performance traits classes was started with C and finished with P. there is no data for some classes. You should mention it at methods section. No 15 classes.

6. Line 265: The legs structure traits classes was started with C and finished with P. there is no data for some classes. You should mention it at methods section. No 15 classes.

7. Line 298: for Table 5, please give the median values of score data.

8. Line 379: “Figure 2. Second Factorial Analysis”. “Factorial” should e “Factor”

Reviewer 2 Report

Comments and Suggestions for Authors

The submitted paper covers a very interesting subject on specific evaluation of an endangered horse breed. Overall, the paper is well-written. However, there are several important corrections and clarifications needed to ensure a well-structured document and facilitate understanding to readers.

The primary issue pertains to the structure of the tables and the definition of traits. Additionally, some sentences require further explanation.

Lines 41-42: Please, review the keywords. Factor analysis and Maremmano horse are also included in the title. Consider replacing them with more informative keywords.

Introduction:

More information about the Maremmano breed should be provided to readers, such as current census data, distribution, number of breeders, main objectives of the breeding program…

Line 83: “In Italy” should be lowercase as “in Italy”.

Material and methods:

More information regarding the animals included in the analysis is necessary, such as their average age, minimum and maximum ages, number of studs, year of data collection, percentage of the population sampled, locations where data were collected, pre-selection criteria, availability of repeated measurements...

Additionally, it is recommended to rename citations of supplementary information with the original titles of the corresponding tables or figures. For instance, in line 106, replace "Additional file 1" with "Figure S1".

A re-structuration of the tables is needed to enhance clarity and ensure easy comprehension for readers. It is suggested to divide table 1 in two separate tables, including the statistical information from table 5. Consequently, table 1 will contain biometric measurements and their statistical parameters, while table 2 will contain continuous evaluated traits and their statistical parameters. Other collected parameters may be omitted if they are supposing for the correct management of data.

Within the tables, it is imperative to maintain the order of the analyzed traits, which may align with the order of traits in figure S1. Additionally, the names of the traits must be consistent across the text, tables and the figure S1.

Perhaps phenotypic correlations between the analyzed traits could be estimated and included into the statistical analysis. However, it is important to note that the trait BSC is not related to legs structure correctness. And a different scale was used for data collection. Hence, we recommend relocating this trait to a different section for clarity and consistency.

Due to the limited number of stallions, readers would benefit from additional information regarding the distribution of their evaluations by age group, judge group and BCS group. It may be helpful to include more detailed information as supplementary material Perhaps you can include more information as supplementary material to provide a comprehensive understanding.

Line 103: Please, correct (40 males and 560 females).

Figure 1.

(1) In the title, replace detected to analyzed.

(2) Review the order of the traits, ensuring consistency across tables and figures, and maintain uniform naming throughout the manuscript. Consider relocating the BSC trait to a different group if necessary.

(3) Modify the name of the legs structure correctness traits to accurately reflect their content. For example: replace “base narrow” with “base narrow – base wide”, replace “toes out” with “toes out-toes in”…

Line 165: specify the number of records available for each animal.

Lines 168-170: review the description of the content of figure S2.

Line 175: For HPL, note that a value of 1 indicates the lowest class. Please, reconsider the scale of this trait and clarify the correct profile for this breed.

Lines 186-187: move this information to the material description section.

Line 189: replace VC to CV

Line 206: please, include more information about the depth of the pedigree.

Line 208: Clarify the criteria used by ANAM for better understanding.

Results:

In the results section, it is very important reference the corresponding table or figure displaying the data to help readers’ access to this information. Any omitted results should be referred as “results not shown”. Additionally, the titles of tables and figures should be enhanced by including more descriptive information to ensure they are concise yet informative

I suggest the use of clearer colors for the figures in order to aid readers the observation and interpretation of the obtained results effectively. It is also important to review the relationship between colors and numbers for consistency. Furthermore, maintaining a consistent order of traits across tables and figures will assist readers in navigating the content more seamlessly.

Table 2, 3 and 4:

(1) Please review the content of the tables. Some frequency data presented as percentages do not total 100%, for example: HPL come to 103%, NLL come to 99%, FFBN come to 102%… If decimals are required, please, include them.

(2) Please, homogenize the format of footnotes according to author guidelines.

(3) You can reference the name of the traits as reference to table 1.

(4) Extreme classes have no observations in all the tables, but this information is only highlighted for table 2.

Line 259: cite traits with values over 80%.

Lines 346-347: you refer to factor 3 as the length factor, but trait NAL is neck alignment which is not a length-related trait.

Lines 351-353: consider moving this sentence to the discussion section.

Lines 367-378: the information about figure 2 is not clearly explained in the text. It would be greatly appreciated if you could improve the explanation.

Figure S1. Include abbreviation of the traits in figure S1, to ensure correspondence with the text. And maintain a consistent order of the traits throughout the manuscript.

Figure S2. consider categorizing this figure as a table for clarity and consistency, and review the color spectrum.

Discussion:

It is very important to include references to the different tables and figures in the discussion section in order to aid readers in following the text.

In general, the discussion is very descriptive. It is necessary to include more critical thoughts about the results and relate them to other studies with similar objectives/methodology/traits.

Line 424: Do you really consider that reducing the number of classes will enhance the use of the entire scale? Have you done any simulation?

Lines 429-431: obtaining evaluations from different judges for the same animal at the same moment is also critical to evaluate the system.

Lines 444-445: Sexual dimorphism is not strong. But the analyzed traits are not clearly related with sexual dimorphism.

Line 448: remove “coefficient of variation” as the abbreviation has been previously used in the main text.

Line 472: replace h2 to heritability. This abbreviation has not been defined before in the main text.

Lines 491-494: please, include the reference of the table where you obtained this information. If it is referred to table 7, the higher value is 0.16 (MS-CG).

Line 496: please, include the range.

Table 7: Consider including more discussion about the Spearman correlation between IBV1 and the other traits in this section.

Conclusion:

Line 521: Consider clarifying the term "totipotent" for readers' understanding

Lines 541-543: Could you explain how you plan to disseminate this information to breeders?

Reference list:

Please, review references 27, 36, 37 and 38 for accuracy and consistency with authors’ guidelines.

Supplementary data:

Figure S1:

(1) please reorder the traits following the same order than tables and figures in main text.

(2) Provide additional clarification for images defining the height of the chest trait. Please review the spelling of this trait.

Figure S2: Consider categorizing this figure as a table. Use clearer colors for better visibility. Delete one side of the duplicated information in the table.

Round 2

Reviewer 1 Report

Comments and Suggestions for Authors

You didn't corrected many of my suggestions.

Author Response

In our opinion an answer has been given to all the questions and only the plausible corrections have been made. However we are available to make further and timely corrections if properly reported

Reviewer 2 Report

Comments and Suggestions for Authors

I would like to thank the authors for their effort in considering the proposals and answering the questions, which have helped me to understand some issues of the work.

General comments are indicated to improve the understanding of the document:

  • Information about census, ages, years of data collection, percentage of the population sampled, locations, preselection, etc., must be included in the final version of the Materials and Methods section. Readers need this information to understand and evaluate the document.

  • The table structure should be reviewed. The structure of Table 5 is unusual. Perhaps the journal's editor can assist with this if you are not available.

  • Figure 1: In the title, replace "detected" with "analyzed." This change has not been made in the new version.

  • Regarding the names of the traits, I understand that you cannot change the official sheet. When I recommended reviewing the names of the traits related to leg structure correctness, it was because you have only included one defect in the name (e.g., base narrow or toe in). However, the review of Supplementary Table 1 shows that the used scale includes two different defects (in the indicated cases: base narrow-base wide and toe in-toe out), and both options are not included in the name of the traits. This omission could mislead readers. A complete description of the traits is very important for a better understanding of the manuscript.

  • In the title of Figure S2, you have not mentioned the inclusion of body measurements, but they are included in the table.

  • Please review if the moved information is actually included in the new position. Some changes are marked as deleted in the new version.

  • The quality of Figure 2 is poor and the colors are not adequate. Please review this comment. It is very important for the final version.

  • Please include your response about NAL and Factor 3 in the final version.

  • Please include your response about Figure 2 in the final version.

  • Please review the comment for lines 491-496. You have added the reference to the table, but the highest value is 0.16 (MS-CG) and you have indicated 0.10 (MS-WH).

  • Please include the clarification of "totipotent" in the final version.

  • Please consider deleting duplicate information in Figure S2. The information is interesting but redundant.

Author Response

Please see the attach (the correction highlighted in yellow were reported in the paper)
